# Unleashing the Power of Generic Segmentation Models: A Simple Baseline for Infrared Small Target Detection

## ABSTRACT

Recent advancements in deep learning have greatly advanced the field of infrared small object detection (IRSTD). Despite their remarkable success, a notable gap persists between these IRSTD methods and generic segmentation approaches in natural image domains. This gap primarily arises from the significant modality differences and the limited availability of infrared data. In this study, we aim to bridge this divergence by investigating the adaptation of generic segmentation models, such as the Segment Anything Model (SAM), to IRSTD tasks. Our investigation reveals that many generic segmentation models can achieve comparable performance to state-of-the-art IRSTD methods. However, their full potential in IRSTD remains untapped. To address this, we propose a simple, lightweight, yet effective baseline model for segmenting small infrared objects. Through appropriate distillation strategies, we empower smaller student models to outperform state-of-the-art methods, even surpassing fine-tuned teacher results. Furthermore, we enhance the model's performance by introducing a novel query design comprising dense and sparse queries to effectively encode multi-scale features. Through extensive experimentation across four popular IRSTD datasets, our model demonstrates significantly improved performance in both accuracy and throughput compared to existing approaches, surpassing SAM and Semantic-SAM by over 14 IoU on NUDT and 4 IoU on IRSTD1k. The source code and models will be released.

## CCS CONCEPTS

• **Computing methodologies** → **Image segmentation**; **Object detection**.

## KEYWORDS

Infrared Small Target Detection, Segmentation, Knowledge Distillation, Segment Anything Model

## 1 INTRODUCTION

Infrared imaging technology offers several advantages over visible light imaging, including robust anti-interference capabilities, adaptability to various environments, and higher discernibility [53, 59, 89]. As a result, it enjoys widespread adoption across various domains such as video surveillance [58, 76], medical and healthcare [15, 27, 28], remote sensing [42, 46, 67] and industrial inspection

[2]. In critical scenarios like ocean rescue or remote sensing, where target scales are small, it is crucial to identify small targets within infrared images. Traditional infrared small target detection (IRSTD) methods fall within the broader spectrum of three specific categories: filter-based [14, 20, 26, 60, 61], local information-based [5, 18, 30, 66], and data structure-based [13, 81, 87].

Recently, deep learning approaches for IRSTD [11, 12, 34, 69, 84, 85] have gained significant attention for their capacity to function without handcrafted priors. However, these data-centric methods pose unique challenges. Constructing a large-scale dataset demands expensive pixel-level annotations while publicly available datasets are often limited in size. Consequently, researchers often resort to data-efficient strategies, such as weakly supervised training [33, 75] or U-shaped models [50] tailored specifically for IRSTD [11, 12, 34, 69, 83–85], departing from architectures [16, 38, 41] commonly used in generic detection and segmentation tasks. Although prior studies have shown that specially designed networks outperform the common architectures in generic tasks, these conclusions often rely solely on training these models from scratch on the small-scale IRSTD dataset, lacking thorough exploration and neglecting resources from visible light images. Notably, the Segment Anything Model (SAM) [31] and its derivatives [29, 36, 71, 80, 90] offer strong backbones trained on extensive datasets and demonstrate effectiveness across various tasks. Thus, it is curious to investigate whether these models offer benefits for IRSTD.

In this study, we aim to build a pioneer model for IRSTD by pre-training on vast visible light data using robust generic segmentation models. This endeavor raises two key questions: 1) How do generic segmentation models like SAM and its derivatives perform in the field of IRSTD? 2) What architectural design effectively facilitates the transferability from these segmentation models to IRSTD? To address the questions, we undertake comprehensive experimentation across various models, including SAM [31], Semantic-SAM [36], SAM-HQ [29], as well as SAM's efficient variants like MobileSAM [80], and EfficientSAM [71]. We compare their performance to established state-of-the-art (SOTA) methods in IRSTD. Despite encountering significant overfitting (Check details in the Appendix) after finetuning, certain SAM-based methods achieve comparable performance to leading IRSTD approaches, as shown in Table 1. Notably, Semantic-SAM consistently outperforms other models. We hypothesize that Semantic-SAM's hierarchical structure enhances its capability to exploit multi-scale features compared to plain transformer architecture. Additionally, its training strategy facilitates the generation of masks with varying granularity, potentially benefiting transferability to the IRSTD task.

Motivated by these findings, we propose to distill the original Swin-based Semantic-SAM encoder into a lightweight backbone to enhance efficiency and transferability while mitigating performance drops from overfitting. Our approach adopts the many-to-many training strategy from Semantic-SAM [36], sharing the decoder

and learning objectives. After pre-training, we replace the decoder with a feature pyramid network (FPN) [38], coupled with a modified SAM decoder to produce high-resolution masks. This refined pipeline yields a simple and lightweight model that surpasses previous IRSTD methods and SAM's efficient variants in performance. Additionally, we introduce a novel query design comprising dense and sparse queries, enhancing model performance through multi-level information fusion. These queries interact with each stage from the encoder to the decoder, ultimately aiding in target prediction. Extensive experiments demonstrate that our model achieves state-of-the-art performance across four public datasets. Remarkably, it achieves a mIoU of 97.0 on the NUDT dataset, underscoring its exceptional capabilities.

In summary, the contributions of this study are as follows:

- We investigate the SAM and its variants in the context of IRSTD through extensive experiments. Our findings reveal their comparable performance with state-of-the-art methods, offering valuable insights into adapting generic segmentation models for IRSTD
- We propose a simple baseline model leveraging generic segmentation models via knowledge distillation. It incorporates novel query designs to effectively encode multi-scale features through interaction with both the encoder and decoder.

## 2 RELATED WORK

### 2.1 IRSTD Methods

IRSTD differs in objective from generic detection tasks. Previous works have often approached IRSTD as a segmentation task, prioritizing this perspective for improved optimization. Dai *et al.* introduce the first public dataset for IRSTD [11], shifting the task from model-driven to data-driven. They proposed a U-shaped network featuring a bottom-up multi-level information aggregation module, enhancing the model's detection capabilities. Some other works introduce model-based IRSTD techniques into the network [82, 84, 85]. Recently, Li *et al.* propose a densely connected U-net [34] and Wu *et al.* propose a U-net in U-net architecture to improve the detection performance further [69].

Although U-shaped networks are highly favored for scenarios with limited data and requiring high-resolution output, such as IRSTD, the size of infrared small target data is often inadequate to meet the increasing demands for model performance. One approach to address this issue is leveraging weak supervision to alleviate annotation burdens [33, 75]. However, a more natural avenue for exploration is bridging the connection between IRSTD and generic segmentation tasks, given the abundance of data available for the latter, which can be orders of magnitude larger than IRSTD datasets. In such a setting, U-shaped networks encounter challenges in handling large volumes of data and knowledge transfer due to their high computational complexity along with network depth and substantial differences with plain or hierarchical networks, which are more commonly applied in general segmentation tasks.

### 2.2 Segment Anything Model

The Segment Anything Model (SAM) [31] stands as a pivotal achievement in the fundamental image segmentation field, having received extensive attention over the past year. SAM has showcased remarkable capabilities in zero-shot transfer learning and boasts versatility across a diverse array of vision tasks. These tasks span a broad spectrum, encompassing medical image analysis [44, 68, 78], detection of camouflaged objects [6, 21, 57], object tracking [10, 72], analysis of AI-Generated Content (AIGC) [56, 86], and various segmentation tasks [64, 74]. Furthermore, subsequent research efforts have delved into addressing specific needs such as high-resolution output [29], semantic understanding [36], and real-time application [71, 80, 88, 90]. An intuitive idea is to investigate the performance of these models, known for their strong generalization capabilities, in the context of IRSTD. This exploration could shed light on the potential applicability of SAM and other generic segmentation models in addressing the unique challenges posed by IRSTD.

### 2.3 Knowledge Distillation in Segmentation

The majority of research in the realm of segmentation emphasizes semantic awareness, aiming to capture inter and intra-class relations by transferring knowledge from teacher models to student models. In class-agnostic segmentation, distillation techniques typically fall into three categories: direct mimic [49], relation-based [40, 62, 79], and generation-based [3, 47, 73] approaches. With the release of SAM and its widespread real-world applications, there has been a growing interest in the practical deployment of SAM, prompting several works to explore distillation techniques to reduce its computational cost. Recognizing the challenge of coupled training between the image encoder and mask decoder, MobileSAM [80] proposes to decouple their optimization processes, employing simple Mean Squared Error (MSE) loss to mimic the behavior of teacher models directly. EfficientSAM [71], on the other hand, adopts masked image modeling, a generation-based method, to distill SAM into a lightweight Vision Transformer (ViT) model. Additionally, other works introduce efficient SAM variants based on different backbones, with many employing direct mimicry by combining MSE loss, Binary Cross-Entropy (BCE) loss, and Dice loss [45]. While our work does not primarily focus on the real-time application of large vision models to the IRSTD task, we employ distillation techniques to achieve more efficient training and establish a simple yet strong baseline for IRSTD.

### 2.4 Query Design

Drawing inspiration from the Global Workspace Theory in cognitive science, Goyal *et al.* [17] proposed the concept of a shared global workspace (learned arrays) for coordinating multiple specialists. Additionally, the PERCEIVER network family [24, 25] employs a latent array to encode implicit information from the input array. Expanding the scope further, similar approaches have been observed in designs such as Involution [35], and VOLO [77]. In these designs, learnable tokens replace original keys, resulting in dynamic affinity matrices. Subsequently, models like QnA [1] and TransNeXt [54] adopt learnable queries for attention calculation within their backbones, demonstrating effectiveness. Moreover, the two-way transformer design utilized by the SAM decoder can also be interpreted as a project and broadcast workspace encoded by learnable tokens, drawing inspiration from models such as DETR [4], and Maskformer [9].

**Figure 1: The pipeline of our model. First, the pre-trained image encoder takes infrared images as input and generates latent feature maps at four scales. These feature maps are passed through an FPN for bottom-up information aggregation. The decoder takes the output of FPN and makes mask predictions. Further, we incorporate a novel query design in our model for better cross-level information propagation.**

Our proposed query design draws inspiration from models like QnA and TransNeXt. It utilizes learnable queries instead of original features for cross-attention knowledge transfer. Similar to DETR and Maskformer, we also leverage sparse queries to generate the final output. However, what sets our design apart is its operation not only within a single mixer layer, as observed in QnA and TransNeXt, but also across multiple levels. Moreover, we integrate both dense and sparse queries to encode multi-scale information, further enhancing detection accuracy.

## 3 METHODOLOGY

### 3.1 Preliminaries

We first review the training strategies employed by variants of SAM [31]. SAM is designed to accommodate flexible segmentation prompts, allowing for various training approaches. Generally, random sampling from labeled training data can be used to generate prompts, driving the end-to-end training of prompt-based mask prediction networks like SAM. SAM-HQ [29] and Efficient-SAM [71] adopt this strategy by sampling mixed types of prompts, including bounding boxes, randomly sampled points, and coarse masks as input. In contrast, by employing Hungarian Matching, Semantic-SAM adopts a multi-choice learning strategy [19, 37], enabling the network to output six different granularity masks for a single prompt. After finetuning these generic segmentation models on IRSTD datasets, as shown in Table 1 and Appendix, we find: 1) the large generic segmentation models such as SAM, Semantic-SAM, and SAM-HQ encounter significant overfitting issues (see Appendix for details); 2) Despite overfitting, Semantic-SAM consistently outperforms SAM and its variants and achieves comparable performance to state-of-the-art IRSTD approaches. According to the experimental results, we conjecture that Semantic-SAM's superior performance in IRSTD transferability stems from its unique training strategy and hierarchical network architecture compared to other SAM variants. We therefore use powerful Semantic-SAM as the teacher model to empower our proposed small models in

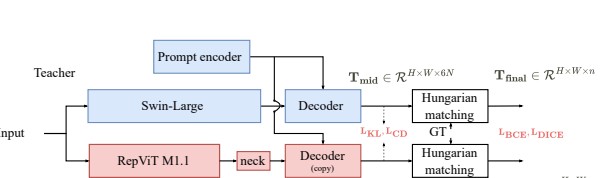

**Figure 2: The proposed distillation framework. The modules in blue are frozen during the distillation process, while the modules in red are trainable.**

IRSTD. The image encoder in the student model is RepViT M1.1 [63] during the pre-training distillation stage and extended to our proposed simple baseline during the fine-tuning stage to align with the different learning objectives. The decoder in the student model is determined by different training stages, which will be illustrated in detail in the following section.

Semantic-SAM comprises three fundamental modules: an image encoder, a prompt encoder, and a mask decoder, akin to SAM and other interactive segmentation models. During training, data is restructured by clustering multiple ground truth (GT) masks of varying levels that share the same click. For each image, $N$ prompts (points or boxes) are sampled. Subsequently, each prompt is linked to six queries through a query-based mask decoder, representing six distinct granularities, resulting in $6 \times N$ output masks. To facilitate multiple predictions matching with GT masks for the same click, Semantic-SAM uses the Hungarian algorithm, enabling many-to-many matching and yielding $n(n \leq 6 \times N)$ final output-GT pairs.

### 3.2 Knowledge Distillation in Pre-training

The backbone in Semantic-SAM, Swin-Large [41], consumes approximately 197 million parameters and 200 GFLOPs when processing 512×512 images. This poses a great challenge for the model's deployment in the real world, especially in edge devices. Besides, such a large model's fine-tuning in infrared target detection (IRSTD)

usually encounters overfitting issues because of the small scale of labeled samples, *i.e.*, several hundred to a thousand samples in IRSTD datasets. To this end, we resort to knowledge distillation to help the proposed lightweight backbone efficiently learn knowledge from the powerful teacher, *i.e.*, Semantic-SAM, while mitigating performance drops from overfitting.

During knowledge distillation, the encoder of the student model is RepViT M1.1 [63], while the decoder is copied from the pre-trained Semantic-SAM decoder as shown in Figure 2. Besides, we add a lightweight neck module following the student backbone to align the channel dimension between the image encoder and decoder. The distillation is conducted on the part of the SA-1B dataset [31]. The model is optimized by minimizing the disparity between the outputs of the student model and those of Semantic-SAM.

Current work for efficient SAM variants only trains the image encoder part during their distillation stages [80], using MSE loss to mimic the teacher encoder's output directly. Despite their success, we find its inadequacy in fully exploiting the rich granularity representation of Semantic-SAM's decoder output, as features from the image encoder do not directly correspond to the final output mask while the decoder's outputs encapsulate much richer task-related information. Hence, as shown in Figure 2, we adopt a combination of binary cross-entropy (BCE) loss and DICE loss [45] in the pre-training stage to align the student's outputs $S_{final}$ with teacher's final outputs $T_{final}$. Technically, we propose to employ KL-divergence loss along both the channel [55] and spatial [23] dimensions between the intermediate teacher and student outputs $T_{mid}, S_{mid}$, *i.e.*, the $6 \times N$ outputs before Hungarian Matching, to help the student recognize the significance of Semantic-SAM's outputs. This combination aims to maintain the shapes of masks and simultaneously highlight the relationships among different granularities, thereby enhancing the distillation performance. The final distillation loss can be formulated as follows:

$$L_{DIS} = L_{BCE} + \lambda * (L_{DICE} + L_{KL} + L_{CD}), \quad (1)$$

where $L_{BCE}, L_{DICE}, L_{KL}$ and $L_{CD}$ represent the BCE loss, DICE loss, vanilla KL loss, and channel-wise KL loss, respectively. $\lambda$ is a hyper-parameter to balance the losses.

## 3.3 Model Design

After pre-training, we take the pre-trained student backbone as the image encoder in our proposed baseline model for IRSTD. We follow EdgeSAM [91] to integrate a tiny FPN behind the image encoder to enhance multi-scale feature representation. Besides, we modify the SAM decoder to handle high-resolution inputs from the FPN. FPN and the new decoder are both re-initialized, which helps the model avoid overfitting issues. Apart from the above design, we introduce a novel query design comprising dense and sparse queries that interact with the image encoder, FPN, and mask decoder, to further enhance the propagation of semantic information and integrate features across various scales.

The popular multi-scale module FPN progressively upsamples the features from the bottom and performs spatial element-wise addition. However, we observe from experiments that the resulting model tends to rely more heavily on the features from the top layers rather than the image encoder's deep layers, which contain rich

**Algorithm 1** Pseudocode of query design in image encoder.

```
# Variables: Encoder queries Q_encoder, Q_dense
# Functions: Image_Encoder()
def init():
    Q_encoder = Embeddings(n,d)
def Sparse_func(Q, S):
    Q, S = Cross_attn(q=Q, k=S, v=S)
    Q = Self_attn(MLP(Q))
    Q, S = Cross_attn(q=S, k=Q, v=Q)
def Dense_func(Q, S):
    list = []
    list.append(Q).append(S)
    Q ,S = Deformable_attn(list)
def forward(I):
    for layer in Image_Encoder():
        S_i = layer(S_i-1)
        Q_dense = Query_embed(S_0)
        Q_encoder, S_i = Sparse_func(Q_encoder, S_i)
        Q_dense, S_i = Dense_func(Q_encoder, S_i)
```

and high-level semantic information. This phenomenon leads to a critical scenario where the clearly discriminative targets depending on the deeper layer features are not recognized as predictions by the decoder, resulting in low detection accuracy. Therefore, we aim to build a more effective multi-level aggregation module that can encode critical information from layer to layer. This module should seamlessly integrate into various architectures and be applicable throughout the network, offering versatility and adaptability. Inspired by [1, 17, 54], we propose a novel design based on query learning to enhance information aggregation and better semantic information propagation.

***Query design:*** As illustrated in the top left part in Figure 1, the proposed design consists of two types of queries: sparse queries and dense queries. For dense queries, we initialize them as $\mathbf{Q}_{dense} \in \mathcal{R}^{m \times \frac{H}{2} \times \frac{W}{2}}$ by a duplication of the image encoder's first stage output. Recognizing the significant computational complexity of cross-attention mechanisms, we opt for multi-scale deformable attention [92] between dense queries and the image features of the image encoder's next three stages. The deformable attention has linear complexity with the spatial size and thus will not introduce much computation burden. For sparse queries, we categorize them into three groups based on their initial interaction points with the model, *i.e.*, sparse encoder queries $\mathbf{Q}_{encoder} \in \mathcal{R}^{n \times d}$, sparse FPN queries $\mathbf{Q}_{FPN} \in \mathcal{R}^{n \times d}$, and sparse decoder query $\mathbf{Q}_{decoder} \in \mathcal{R}^{1 \times d}$, where $n$ is the number of queries (4 by default) and $d$ is the dimension size. All the sparse queries are learnable and initialized from scratch and have the same channel dimension size. Note that $\mathbf{Q}_{decoder}$ only has one query corresponding to the final output mask. As illustrated in the top right part of Figure 1, within the image encoder, the sparse encoder queries interact with the features of the image encoder's four stages in a bottom-up fashion, and each interaction is achieved by bi-direction attention follows by four steps: (1) cross-attention from queries to features, (2) a point-wise MLP to encode the queries, (3) self-attention on queries, (4) cross-attention from features to queries. The output of steps 3 and 4 are the updated sparse queries and image features for the following modules. Then, the obtained sparse encoder queries concatenate with the next sparse tokens, *i.e.*, FPN queries, and then interact with each granularity level of features within FPN through several bi-direction attention operations

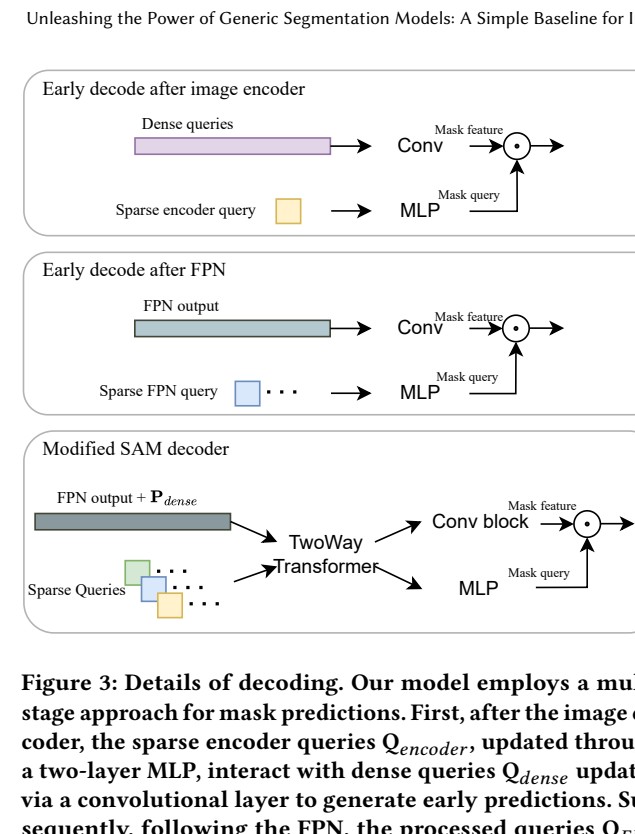

**Figure 3: Details of decoding. Our model employs a multi-stage approach for mask predictions. First, after the image encoder, the sparse encoder queries $\mathbf{Q}_{encoder}$, updated through a two-layer MLP, interact with dense queries $\mathbf{Q}_{dense}$ updated via a convolutional layer to generate early predictions. Subsequently, following the FPN, the processed queries $\mathbf{Q}_{FPN}$ are combined with the FPN output to produce intermediate predictions. In the final stage, the $\mathbf{Q}_{encoder}$, $\mathbf{Q}_{FPN}$ and $\mathbf{Q}_{decoder}$ are incorporated into the modified SAM decoder. After interacting with image features through a two-way transformer, $\mathbf{Q}_{encoder}$ and $\mathbf{Q}_{FPN}$ are discarded, and the decoder makes mask predictions with a spatially point-wise product between mask features and $\mathbf{Q}_{decoder}$ updated by MLP.**

in a top-down manner. Note that all levels of FPN features have the same channel dimension size, guaranteeing dimension consistency between sparse queries and FPN features. Finally, all sparse queries are concatenated together, and useful information from features within the decoder is obtained through bidirectional attention between queries and features. We summarize the pipeline of our query design in the image encoder as pseudocode in Algorithm 1.

**Decoding process:** As illustrated in Figure 3, the model involves three decoding processes throughout the entire pipeline, *i.e.*, two early decoding processes and one final decoding process. First, after the image encoder, we apply a convolutional layer to the dense queries $\mathbf{Q}_{dense}$ as the mask feature and feed the first sparse encoder query $\mathbf{Q}_{encoder} \in \mathcal{R}^{1 \times d}$ to a 2-layer MLP simultaneously, resulting in a mask prediction by spatially point-wise product between the mask feature and the MLP's output. The process after FPN is similar. We use FPN output as mask features and the first sparse FPN query $\mathbf{Q}_{encoder} \in \mathcal{R}^{1 \times d}$ as the other multiplier. The early mask prediction after FPN is encoded by a lightweight convolutional block and then added back to FPN feature maps as clues, following the procedure of dense prompt in SAM. We observe from experiments the

early decoding processes facilitate effective information propagation between different modules, further enhancing the mask quality predicted by the final decoder. For the final decoding process, we modified the SAM's decoder by replacing the 2-layer deconvolutional layer with a two-layer $3 \times 3$ convolutional block, since we already have high-resolution features from the hierarchical architecture. Several stacked two-way transformer blocks process the sparse queries and the image feature maps. Then the dot product between the sparse decoder query and feature maps constructs the final mask prediction. The overall process can be formulated as:

$$\mathbf{Z} = \mathbf{ImageEncoder}(\mathbf{I}, \mathbf{Q}_{encoder}), \quad (2)$$

$$\mathbf{F} = \mathbf{FPN}(\mathbf{Z}, \mathbf{Q}_{encoder}, \mathbf{Q}_{FPN}), \quad (3)$$

$$\mathbf{M} = \mathbf{Decoder}(\mathbf{F}, \mathbf{Q}_{encoder}, \mathbf{Q}_{FPN}, \mathbf{Q}_{decoder}), \quad (4)$$

where $\mathbf{I}$, $\mathbf{Z}$, and $\mathbf{M}$ denote the input images, feature maps after encoder, and mask prediction, respectively.

## 4 EXPERIMENTS

### 4.1 Experimental Settings

***Datasets***. During the distillation process, we conduct training on 1% of the entire SA-1B dataset with files named from sa_000000 to sa_000009. We monitor the distillation pre-training progress using the evaluation set of COCO2017 [39] with panoptic segmentation annotations.

To evaluate our methods in the context of IRSTD, we consider four publicly available datasets: SIRST [11], NUDT [34], IRSTD1k [85], and MDFA [65]. The SIRST dataset contains 420 infrared images with resolution varying from $100 \times 100$ to $300 \times 300$. We follow [11] to split 256 images as training set, and the rest are for evaluation set. The NUDT dataset proposed in [34] contains 1,327 $256 \times 256$ images and we adhere to their approach by assigning 663 images to the training set and the remaining images to the evaluation set. IRSTD-1k dataset provides 1,001 images at the resolution of $512 \times 512$. Following [85], we select 800 images as the training set. Notably, we exclude six images from the remaining set due to inaccurate annotations. To ensure fairness, we test all methods under the same settings and provide details of these excluded images in the appendix. Additionally, the MDFA dataset comprises 10,000 images for the training set and 100 images for the evaluation set.

***Network details***. The RepViT [63] is a hierarchical model that outputs latent features of four different sizes: $\{\frac{1}{4}, \frac{1}{8}, \frac{1}{16}, \frac{1}{32}\}$. In the context of the IRSTD task, we observe that the large downsampling rate in the original backbone is too aggressive for detecting tiny targets. Therefore, we adjust the initial embedding of RepViT from $4\times$ downsampling to $2\times$ downsampling for IRSTD1k and $1\times$ for the other three datasets. For the tiny FPN employed after the image encoder, it first applies 4 convolutional layers with $1 \times 1$ kernel size to map the output from the image encoder uniformly to 256 channels. Then, the smaller-size feature maps are upsampled through nearest interpolation and added to larger feature maps for multi-level information aggregation. Finally, a $3 \times 3$ convolutional layer is employed to process the output. For the proposed query design, we set the number of queries $n$ 4 for $\mathbf{Q}_{encoder}$, 4 for $\mathbf{Q}_{FPN}$ and 1 for $\mathbf{Q}_{decoder}$. The $\mathbf{Q}_{dense}$ are duplicated from the image encoder's first stage output. The architecture design and hyper-parameters of

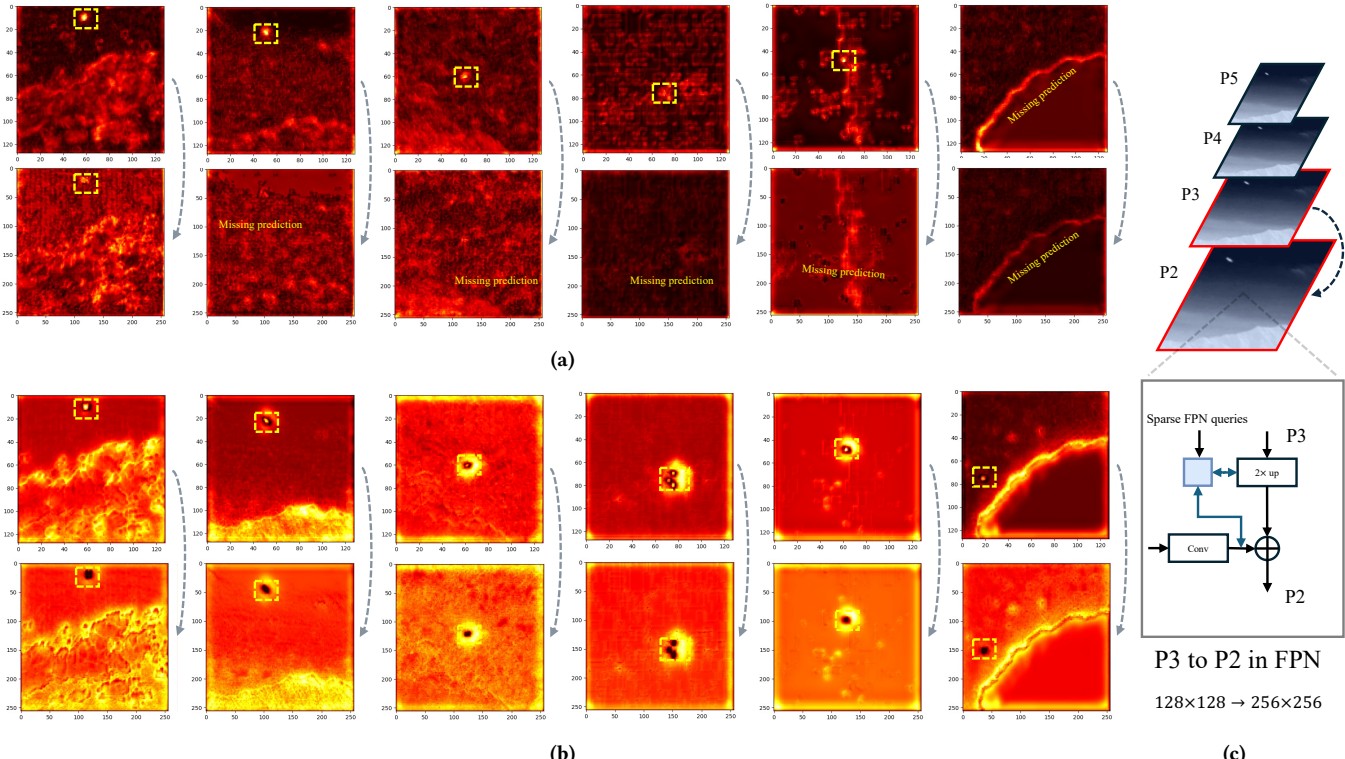

**Figure 4: The ablation study on the proposed query design, From the left to right is (a) Heatmaps of P3 and P2 stages before learned queries are applied. (b) Heatmaps of P3 and P2 stages after learned queries applied. (c) Specific location of P3 and P2 in FPN.**

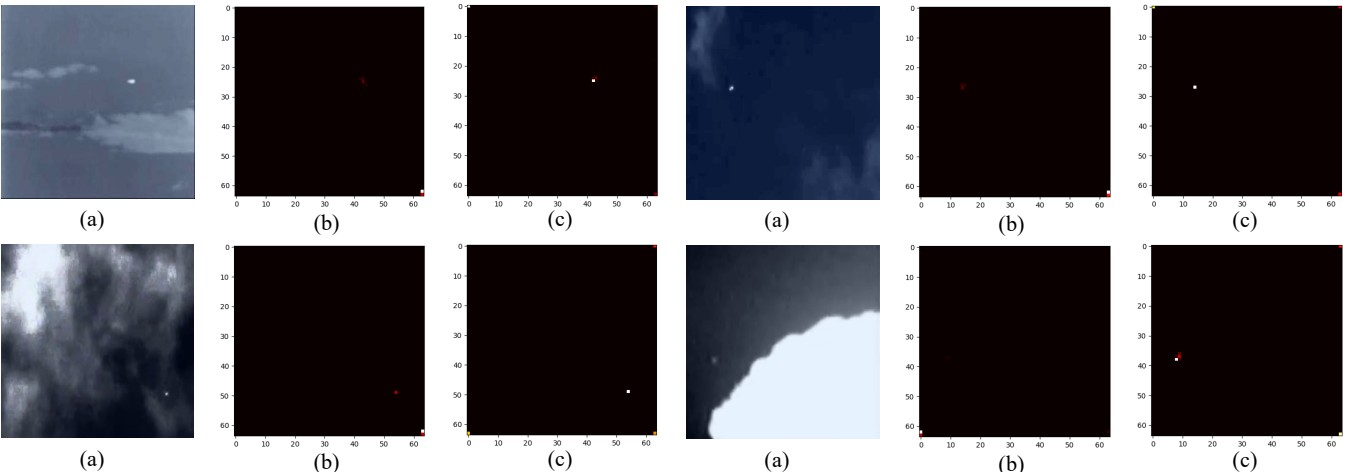

**Figure 5: Ablation study on early decoding through visualization of attention maps where the main branch features attend to the queries. (a), the input images. (b), the attention maps without early decoding. (c), the attention maps with early decoding.**

**Table 1: A comprehensive comparison with previous IRSTD approaches and generic segmentation models on the NUDT, IRSTD1k, SIRST and MDFA datasets. The evaluation metrics are IoU ($10^{-2}$), $P_d$ ($10^{-2}$) and $F_a$ ($10^{-6}$), the best results are highlighted.**

| Method | Publication | Type | NUDT | | | IRSTD1k | | | SIRST | | | MDFA | | |
|---|---|---|---|---|---|---|---|---|---|---|---|---|---|---|
| | | | IoU ↑ | $P_d$ ↑ | $F_a$ ↓ | IoU ↑ | $P_d$ ↑ | $F_a$ ↓ | IoU ↑ | $P_d$ ↑ | $F_a$ ↓ | IoU ↑ | $P_d$ ↑ | $F_a$ ↓ |
| ACM [11] | WACV'21 | | 68.90 | 97.05 | 11.29 | 62.41 | 91.44 | 35.58 | 70.77 | 93.08 | 3.7 | 40.83 | 83.08 | 90.33 |
| FC3-Net [84] | ACM MM'22 | | 78.56 | 93.86 | 23.922 | 65.07 | 91.54 | 15.55 | 72.44 | 98.14 | 10.85 | 45.62 | **85.29** | 56.76 |
| ISNet [85] | CVPR'22 | Specific | 81.77 | 96.3 | 44.47 | 69.93 | 92.6 | 9.21 | 79.83 | 99.02 | 4.61 | 43.44 | 76.42 | 238.15 |
| DNA-net [34] | TIP'23 | | 88.99 | 98.62 | 4.7798 | 69.38 | 93.3 | 11.66 | 79.26 | 98.48 | 2.3 | 41.44 | 75.73 | 180.66 |
| UIU-net [69] | TIP'23 | | 92.19 | 97.77 | 15.44 | 69.96 | 91.54 | 65.93 | 70.13 | 95.37 | 35.36 | 41.28 | 75.73 | 86.66 |
| SAM [31] | ICCV'23 | | 74.10 | 98.3 | 13.32 | 69.12 | 92.61 | 5.88 | 75.21 | 99.07 | 6.82 | 45.27 | 83.08 | **14.64** |
| SAM-HQ [29] | NIPS'23 | | 74.02 | 98.31 | 14.48 | 68.85 | 91.54 | 9.56 | 75.27 | 97.22 | 2.87 | 44.99 | 81.61 | 24.41 |
| Efficient-SAM [71] | CVPR'24 | Generic | 63.20 | 93.75 | 19.51 | 68.29 | 91.24 | 11.58 | 71.57 | 98.14 | 5.744 | 41.9 | 76.47 | 77.51 |
| MobileSAM [80] | Arxiv'23 | | 59.91 | 96.61 | 19.39 | 65.37 | 88.73 | 10.28 | 64.96 | 97.22 | 12.74 | 33.84 | 67.64 | 150.14 |
| Semantic-SAM [36] | Arxiv'23 | | 83.18 | 97.14 | 12.36 | 70.27 | 92.25 | 20.16 | 78.67 | 99.07 | 5.48 | 45.53 | 0.8 | 273.85 |
| Ours | | | 95.53 | 99.15 | 9.07 | 71.28 | 92.25 | 11.89 | 74.49 | 96.29 | 29.97 | 43.74 | 78.67 | 23.19 |
| Ours+query design | | | **97.04** | **99.55** | 0.6897 | **74.21** | **94.36** | 6.47 | **79.83** | **100** | 2.05 | **46.86** | 83.08 | 24.41 |

**Table 2: Ablation study of the key modules in our model. We show a roadmap for transforming the baseline model to our final model step by step. To better investigate the impact of each component, we highlight the gain in red and degradation in blue.**

| step | Method | NUDT | | | IRSTD1k | | | SIRST | | | MDFA | | |
|---|---|---|---|---|---|---|---|---|---|---|---|---|---|
| | | IoU | $P_d$ | $F_a$ | IoU | $P_d$ | $F_a$ | IoU | $P_d$ | $F_a$ | IoU | $P_d$ | $F_a$ |
| 0 | Baseline model | 89.59 | 98.62 | 35.27 | 66.41 | 90.49 | 17.74 | 60.77 | 95.37 | 107.35 | 41.9 | 89.7 | 115.35 |
| 1 | +Distillation | 95.53 (+5.94) | 99.15 (+0.53) | 9.07 (+26.2) | 71.28 (+4.87) | 92.25 (+1.76) | 11.89 (+5.85) | 74.49 (+13.72) | 96.29 (+0.92) | 29.97 (+77.38) | 43.74 (+1.84) | 78.67 (-11.03) | 23.19 (+92.16) |
| 2 | +Query design in FPN | 95.22 (-0.31) | 99.36 (+0.21) | 8.8 (-0.27) | 71.28 (+0.0) | 92.95 (+0.70) | 11.58 (+0.31) | 74.5 (+0.01) | 97.22 (+0.93) | 17.95 (+12.02) | 43.38 (-0.36) | 84.55 (+5.88) | 32.34 (-9.15) |
| 3 | +Early decoding after FPN | 96.14 (+0.92) | 99.36 (+0.00) | 4.13 (+4.67) | 71.69 (+0.41) | 93.3 (+0.05) | 10.93 (+0.65) | 75.58 (+1.08) | 99.07 (+1.85) | 24.23 (-6.28) | 45.04 (+1.66) | 81.61 (-2.94) | 18.54 (+13.80) |
| 4 | + Extending query design to image encoder | 92.57 (-3.57) | 98.94 (-0.42) | 5.58 (-1.45) | 72.23 (+0.54) | 93.36 (+0.06) | 9.91 (+1.02) | 76.43 (+0.85) | 100 (+0.93) | 15.98 (+8.25) | 45.26 (+0.22) | 83.28 (+1.67) | 24.41 (-5.87) |
| 6 | +Early decoding after image encoder | 96.46 (+3.89) | 99.36 (+0.42) | 1.81 (+3.77) | 73.68 (+1.45) | 93.66 (+0.30) | 7.43 (+2.48) | 77.99 (+1.56) | 100 (+0.00) | 7.97 (+8.01) | 46.09 (+0.83) | 86.76 (+3.48) | 39.06 (-14.65) |
| 7 | +Queries and early prediction as prompt | 97.04 (+0.58) | 99.55 (+0.19) | 0.69 (+1.12) | 74.21 (+0.53) | 94.36 (+0.70) | 6.47 (+0.96) | 79.83 (+1.84) | 100 (+0.00) | 2.05 (-5.92) | 46.86 (+0.77) | 83.08 (-3.68) | 24.41 (+14.65) |

the decoder are consistent with SAM's decoder except for replacing the upsample block with a two-layer $3 \times 3$ convolutional block.

**Pre-training details.** During pertaining on SA-1B, we adopt distillation loss $L_{DIS}$ mentioned in section 3.2, and the hyper-parameter $\lambda$ is set to 5. Then, we train the model for 20 epochs using the PyTorch framework with a batch size of 16. Following [36], we use AdamW optimizer[43] with a multi-step learning rate. Initially, the learning rate is set to 1e-4 and reduced by 10 at 90% and 95% of the total number of steps. The training process is conducted on 8 Nvidia GeForce 4090 GPUs.

**Tuning details on IRSTD datasets.** We use a combination of binary cross entropy loss and DICE loss [45] for the fine-tuning stage: $\mathbf{L}_{mask} = \mathbf{L}_{BCE} + \lambda_{DICE}\mathbf{L}_{DICE}$, where $\lambda_{DICE}$ is set to 5. Additionally, we follow PointRend [32] and Implicit PointRend [8], which demonstrate that segmentation models can effectively train with their mask loss calculated using a subset of randomly sampled points instead of the entire mask.

After resizing images from the SIRST dataset to $256 \times 256$, we acquire four datasets with three different sizes: IRSTD1k with sizes of $512^2$, SIRST and NUDT with $256^2$, and MDFA with $128^2$. Then, we train our model for 150 epochs with a cosine learning rate schedule from 1e-4 to 1e-6 with 10 warm-up iterations. For data augmentation, we use a random resize (uniformly from 0.5 to 2.0) and fixed-size crop from Detectron 2 [70]. Notably, we do not apply data augmentation on the NUDT dataset, as we have observed a degradation in performance.

**Evaluation metrics.** Following previous works [11, 33, 69, 84, 85], we adopt the intersection of union ($IoU$), probability of detection ($P_d$), and false-alarm rate ($F_a$) as evaluation metrics.

**Baselines.** To demonstrate the effectiveness of our model, we select five state-of-the-art IRSTD methods for comparison. Since these models are not trained on the SA-1B dataset, we include three large vision models SAM [31], SAM-HQ [29] and Semantic-SAM [36], as well as two efficient variants of SAM: MobileSAM [80] and EfficientSAM [71], for a comprehensive comparison.

Specifically, SAM is trained on the SA-1B dataset for approximately 2 epochs, starting from a pre-trained ViT model. Semantic-SAM is trained using seven datasets, *i.e.*, SA-1B, COCO panoptic [39], ADE20k panoptic [93], PASCAL part [7], PACO [48], PartImageNet [22], and Objects365 [52]. SAM-HQ fine-tunes the pre-trained SAM model on a high-quality dataset, HQSeg-44K [29]. Regarding the efficient variants, MobileSAM is trained on 1% of the SA-1B dataset, similar to our approach. EfficientSAM is initially pre-trained on the ImageNet-1K training set [51] and then fine-tuned on the entire SA-1B dataset.

## 4.2 Main Results

We conduct comprehensive experiments involving five state-of-the-art IRSTD approaches and five generalist segmentation models on four datasets, as summarized in Table 1. Our model demonstrates strong performance across different datasets and scales. On the IRSTD1k dataset with an image size of $512^2$, our model outperforms the second-best model, Semantic-SAM, by approximately 4 IoU, achieving the highest detection probability of 94.36% while

maintaining the lowest false-alarm rate $F_a$. On the NUDT and SIRST datasets with image sizes of $256^2$, our model achieves an impressive 97.04 IoU, 99.55% detection probability, and 0.6897e-4% false-alarm rate on the NUDT dataset, and 79.83 IoU, 100% detection probability, and 2.05e-4% false-alarm rate on the SIRST dataset. Regarding the MDFA dataset with an image size of $128^2$, our model still delivers robust performance, reaching 46.86 IoU, 83.08% detection probability, and 24.41e-4% false-alarm rate.

## 4.3 Ablation Study

*4.3.1 The Journey to Our Model.* As shown in Table 2, our journey begins with a baseline model consisting of three components: RepViT M1.1 as the image encoder, followed by an FPN and a modified SAM decoder. Without distillation and learned queries, the model demonstrates subpar performance across all four datasets.

Subsequently, we conduct knowledge distillation from Semantic-SAM using 1% of the SA-1B datasets, as outlined in step 1. This process incorporates three essential factors: the multi-granularities awareness from Semantic-SAM and the multi-choice training strategy employed by Semantic-SAM, together with abundant segmentation priors derived from visible images. This effort substantially enhances the model's performance, resulting in significant improvements of 5.94, 4.87, 13.72, and 1.84 IoU on the NUDT, IRSTD1k, SIRST, and MDFA datasets, respectively. This establishes a strong model that outperforms previous state-of-the-art IRSTD methods and SAM variants.

Then, to address the ineffectiveness of FPN, we introduce a novel query design to levitate multi-scale information, as outlined from step 2 to step 3 in Table 2, and extend it to the image encoder in the stage of step 4 and step 5. Furthermore, we propose to use sparse queries and early predictions to prompt the decoder, as noted in step 7. Our final model significantly outperforms the pre-trained model, achieving a gain of 1.51, 2.93, 5.34, and 3.12 IoU, as well as improvement in detection probability of 0.4%, 2.11%, 5.34%, and 3.12% on the NUDT, IRSTD1k, SIRST, and MDFA datasets, respectively. Furthermore, we observed a reduction in false-alarm rates from 9.07e-4% to 0.69 e-4%, from 11.89e-4% to 6.47e-4%, and from 29.97e-4% to 2.05e-4% on the NUDT, IRSTD1k, and SIRST datasets, respectively. These results validate the effectiveness of the key designs in our model for enhancing detection performance.

*4.3.2 Analysis on the Query Design.* As illustrated in 4.3 and Table 2, introducing the query design significantly enhances the model's performance. Here, We further analyze the impact of the learned queries by visualizing specific layers and attention maps in Figure 4 and Figure 5.

In particular, we visualize the output of the P3 and P4 stages before and after the queries are applied, as depicted in Figure 4c. The heatmaps in Figure 4a and 4b highlight the differences. Our finding indicates that within the vanilla FPN, the targets identified by higher-level feature maps are diminished after the fusion with low-level features. However, this issue is substantially alleviated by the proposed queries. In Figure 4b, we consistently observe clearer expression of targets in both the P3 and P2 stages. The resulting output demonstrates improved visual quality with finer-grained edges, which is achieved through the combination of high-level semantics contained in queries and high-resolution feature maps.

**Study on the impact of early decoding**. The direct training of queries, by linking them to the output prediction, plays a pivotal role in enabling queries to retain valuable information. As demonstrated in Table 2, the removal of direct training of queries leads to a significant deterioration in performance, evidenced by a noticeable decline in IoU points on the NUDT, IRSTD1k, SIRST, and MDFA datasets. Specifically, there is a drop of 3.89, 1.45, 1.56, and 0.83 IoU points on these datasets.

In Figure 5, we visualize the attention map where the queries attend to the main branch features. Specifically, Figure 5 (a) shows the input images, while Figure 5 (b) and (c) depicts the attention maps without and with early decoding, respectively. The attention maps in Figure 5 (c) exhibit a more accurate response to the target compared to those in Figure 5 (b), underscoring the importance of early decoding.

**Computational complexity analysis**. Our proposed query design strikes a balance between quality and efficiency. Given an input $x \in \mathcal{R}^{b \times h \times w \times d}$ and sparse queries such as encoder queries $\mathbf{Q}_{encoder} \in \mathcal{R}^{b \times n \times d}$ where $b$ is the batch size, $h$, $w$, and $d$ denote the height, width, and dimension of the input, $n$ is the number of queries. For a bi-direction attention module depicted in Figure 1, the computational complexity is:

$$O_{bi-attn.} = 34bnd^2 + 8bhwd^2 + 8bnhwd + 4bn^2d \quad (5)$$

Here, we consider the impact of linear projection and dot product for the complexity above. Since $n$ is set to 4 for $\mathbf{Q}_{encoder}$ and $\mathbf{Q}_{FPN}$, $h \times w \gg d$, the complexity is dominated by the second term. The module is of linear complexity with spatial size.

The multi-scale deformable attention module involving dense queries $\mathbf{Q}_{dense}$ is also of linear complexity with $h$ and $w$. The details can be checked in [92]

## 5 CONCLUSION

This paper presents a robust segmentation baseline for Infrared Small Target Detection (IRSTD). We begin by investigating the capabilities of the popular vision foundation model SAM and its variants in the context of IRSTD. Subsequently, we propose to use a specific distillation strategy to transfer knowledge from generic models to a more efficient architecture, thus establishing a simple, efficient, yet effective baseline, unleashing the potential of the generic segmentation models. Based on the pre-trained model, we introduce a novel query design to aggregate multi-level features and facilitate effective cross-level semantics propagation. Extensive experiments conducted on four public IRSTD datasets showcase the significantly improved performance of our model compared to SAM, its variants, and previous state-of-the-art methods in IRSTD.

**Limitations**. Although we demonstrate that a large amount of visible light data can benefit the IRSTD, training on such data requires considerable time and resources. We encourage future research to delve deeper into analyzing the impact of the type and quantity of visible light images on infrared detection ability. By conducting thorough analyses, researchers can identify the most effective strategies for training more efficiently. This could lead to simpler and more effective approaches for IRSTD, ultimately benefiting various applications and domains.

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
