# OpenReview forum: "Unleashing the Power of Generic Segmentation Model: A Simple Baseline for Infrared Small Target Detection"
_acmmm.org/ACMMM/2024/Conference — MM2024 Poster_

### Official Review · Reviewer_hYKi · 2024-05-12

**Rating:** 5
**Confidence:** 2

**Summary:**

This paper investigated the adaptation of generic segmentation models for the infrared small target detection task. On the one hand, this investigation demonstrates that existing generic segmentation models can achieve comparable performance to specific methods in infrared small target detection tasks. On the other hand, this work also focuses on the untapped potential within IRSTD and develops a simple, lightweight, and effective baseline model using techniques such as knowledge distillation and query design. Extensive experiments demonstrate the SOTA performance of the proposed model.

**Strengths:**

1. These experimental results are convincing. This manuscript is standardized and the writing is fluent.
2. The research motivation is clear and meaningful.

**Limitations:**

1. The pseudocode in Algorithm 1 lacks the necessary annotations.
2. The results in Figure 5 should be zoomed in locally for better observation.
3. For evaluation metrics, the details of the probability of detection (Pd) and false alarm rate (Fa) metrics are lacking.
4. Where can the qualitative results corresponding to Table 1 be found?

**Suitability:**

3

---

### Official Review · Reviewer_oPNE · 2024-05-22

**Rating:** 3
**Confidence:** 4

**Summary:**

The paper proposes a new method for infrared small target detection (IRSTD) by adapting the Segment Anything Model (SAM). The study reveals that generic segmentation models can perform comparably to state-of-the-art IRSTD methods. The proposed method includes a novel query design and knowledge distillation strategies. Extensive experiments across four IRSTD datasets demonstrate that the new model improves performance in terms of accuracy.

**Strengths:**

1.The introduction of a novel query design for better cross-level information propagation enhances the model's ability to detect small targets in infrared images.

2.The use of knowledge distillation to transfer capabilities from larger models to a more efficient architecture is well-executed, leading to improved performance without substantial computational overhead.

3.The paper provides extensive experimental results on four popular IRSTD datasets, showing significant performance improvements over existing methods.

**Limitations:**

1.While the authors propose some improvements, many techniques (like FPN, knowledge distillation, and query design) are widely used in other fields. The paper's uniqueness in terms of technical innovation might not be very prominent.

2.The authors claim in the introduction that they propose a "simple and lightweight model",  there is no relevant data or evidence provided later in the paper to support this claim. This makes it difficult for readers to assess the actual lightweight nature of the model and its performance in terms of resource consumption and efficiency.

3.In the experimental section, the authors do not use the same experimental settings for different large models. This inconsistency raises questions about the fairness and comparability of the results. Could the authors provide a rationale for not using the same experimental settings for all the models?

4.The tables in the paper are not well-designed, which affects the clarity and readability of the presented data. Better visualization techniques could improve the comprehension and impact of the results.

**Suitability:**

3

---

### Official Review · Reviewer_Y3VF · 2024-05-26

**Rating:** 4
**Confidence:** 2

**Summary:**

This paper proposes a segmentation baseline for infrared small target detection. It designs a distillation strategy for knowledge transfer from generic models to a more lightweight model and introduce a query design for hierarchical information aggregation. Experiments on multiple datasets demonstrate the effectiveness of the proposed method.

**Strengths:**

1. The idea of this paper is simple and clearly illustrated.
2. The paper is easy to follow.
3. Both qualitative and quantitative results are provided to illustrate the advantages of the proposed method.

**Limitations:**

1. The cross-modal learning is meaningful for infrared small object detection by means of the potential power of generic segmentation model (SAM). However, the technique contribution of this paper seems not significant. More insights should be presented.
2. Since this work proposes a knowledge distillation strategy, the computation complexity, parameters, and inference expenses should be provided to further demonstrate the advantages.
3. How about the influence of the loss hyperparameter \lambda? The analysis should be provided.

**Suitability:**

3

---

### Meta-Review · Area_Chair_oE8b · 2024-07-02

**Recommendation:** Accept (Poster)
**Confidence:** 3

**Metareview:**

The paper receives mixed reviews of 2 positive and 1 negative ratings. After carefully reading the reviews and the paper, the AC believes this work presents a novel segmentation baseline for infrared small target detection. Moreover, further experimental data indicates that the proposed method achieves good efficiency. Although the reviewers shows some concerns regarding the experiment settings and visualizations, the authors promised to address these issues in the camera-ready paper, given the evidence provided in the rebuttal. The AC leans towards an acceptance.

---

### Meta-Review · Senior_Area_Chairs · 2024-07-10

**Recommendation:** Accept (Poster)
**Confidence:** 4

**Metareview:**

This paper received mixed ratings initially. After rebuttal, two reviewers tend to accept the paper while one still questioned paper quality. SAC and AC carefully checked the paper, reviews and rebuttal and recommend acceptance of the paper.